# Cytomegalovirus-Specific Hyperimmune Immunoglobulin Administration for Secondary Prevention after First-Trimester Maternal Primary Infection: A 13-Year Single-Center Cohort Study

**DOI:** 10.3390/v16081241

**Published:** 2024-08-02

**Authors:** Emmanouil Karofylakis, Konstantinos Thomas, Dimitra Kavatha, Lamprini Galani, Sotirios Tsiodras, Helen Giamarellou, Vassiliki Papaevangelou, Anastasia Antoniadou

**Affiliations:** 1Fourth Department of Internal Medicine, University General Hospital Attikon, National and Kapodistrian University of Athens, 12462 Athens, Greece; costas_thomas@yahoo.com (K.T.); dimitra.kavatha@gmail.com (D.K.); sotirios.tsiodras@gmail.com (S.T.); ananto@med.uoa.gr (A.A.); 2Department of Infectious Diseases, Addenbrooke’s Hospital, Cambridge University Hospitals NHS Foundation Trust, Cambridge CB2 0QQ, UK; 31st Department of Internal Medicine-Infectious Diseases, Hygeia General Hospital, 4 Erythrou Stavrou & Kifisias, Marousi, 15123 Athens, Greece; l.galani@hygeia.gr (L.G.); e.giamarellou@hygeia.gr (H.G.); 4Third Department of Pediatrics, University General Hospital Attikon, National and Kapodistrian University of Athens, 12462 Athens, Greece; vpapaev@gmail.com

**Keywords:** congenital cytomegalovirus infection, immunoglobulin, pregnancy

## Abstract

Primary cytomegalovirus infection during pregnancy has a high risk of vertical transmission, with severe fetal sequelae mainly associated with first-trimester infections. We conducted a retrospective analysis of 200 IU/kg cytomegalovirus-specific hyperimmune globulin (HIG), used in first-trimester maternal primary infections for congenital infection prevention. The primary outcome was vertical transmission, defined as neonatal viruria or positive amniocentesis if pregnancy was discontinued. HIG, initially administered monthly and since 2019 biweekly, was discontinued in negative amniocentesis cases. Women declining amniocentesis and positive amniocentesis cases with normal sonography were offered monthly HIG until delivery as a treatment strategy. The total transmission rate was 29.9% (32/107; 10 pregnancy terminations with positive amniocentesis, 18 completed pregnancies with positive amniocentesis and 4 declining amniocentesis). Maternal viremia was the only factor associated with fetal transmission (OR 4.62, 95% CI 1.55–13.74). The transmission rate was not significantly different whether HIG was started during the first or second trimester (28.2% vs. 33.3%; *p =* 0.58), or between monthly and biweekly subgroups (25.7% vs. 37.8%, *p =* 0.193). Pre-treatment maternal viremia could inform decisions as a predictor of congenital infection.

## 1. Introduction

Congenital cytomegalovirus (CMV) infection (congenital CMV, cCMV) is a major nonhereditary cause of sensorineural hearing loss, and has been associated with neurodevelopmental impairment [1]. cCMV accounts for up to one-fifth of all hearing loss at birth, occurring in 50% and 15% of symptomatic and asymptomatic congenital infections, respectively [2]. The estimated worldwide-pooled birth prevalence of cCMV is 0.67%, with high-income countries having a 3-fold lower infection burden compared to low-and middle-income countries [3].

Vertical transmission can occur by the transplacental passage of maternal bloodborne CMV leading to in utero fetal infection. An absence of prior CMV immunity leads to maternal primary infection, whereas reinfection with a different viral strain or the reactivation of latent maternal CMV infection leads to a nonprimary infection despite the presence of CMV-specific immunity. The presence of preconception immunity confers significant protection against congenital infection in nonprimary compared to primary infections (incidence of vertical transmission 0.15–2% vs. 40%) [4,5].

The risk of intrauterine transmission largely depends on the timing of maternal primary infection, with rates of 36.8%, 40.3% and 66.2% in the first, second, and third trimester, respectively [6]. However, severe sequelae are inversely associated with gestational age, with the highest rates of fetal impairment reported when seroconversion occurs in the periconception period and the first trimester, and virtually non-existent severe cases when infection occurs after the first 14 weeks of gestation [6,7]. Accordingly, prenatal interventions for the secondary prevention of cCMV infection have focused on first-trimester primary infections.

Passive immunization with CMV-specific hyperimmune globulin (HIG) has been used for over two decades as a means of secondary prevention. Observational and randomized studies have described discordant results, while using different dosage regimens and intervals of administration.

The aim of this study was to present a 13-year experience of CMV-specific HIG as a secondary prevention in first-trimester maternal primary CMV infections. We assessed the impact of HIG in reducing cCMV rates and explored potential modifiable and unmodifiable factors associated with vertical transmission. 

## 2. Materials and Methods

This is a retrospective observational cohort study including real-world data of HIG administration in pregnant women with a primary CMV infection diagnosed during the first trimester (≤14th gestational week) [8]. Consultation, treatment and follow-up were carried out at a specialized outpatient clinic in Attikon University Hospital in Athens from 2009 to 2022. The study protocol was approved by the local Scientific Council and Bioethics Committee. Data collection and assessment was conducted by a team of adult and pediatric infectious diseases specialists.

The primary outcome was the vertical transmission of CMV, defined as a positive polymerase chain reaction for CMV deoxyribonucleic acid (DNA) in the newborns’ urine, or in amniotic fluid if pregnancy was discontinued and prior amniocentesis was available. Maternal primary CMV infection was diagnosed in cases of the documented seroconversion of CMV-specific immunoglobulin (G). When both IgG and immunoglobulin (M) were positive, a low IgG avidity titer confirmed a recent primary infection [9]. Laboratory diagnosis was performed at the Pasteur Institute of Athens, a national reference laboratory for CMV serology and molecular diagnosis. Serologic testing was conducted through enzyme-linked immunosorbent assay (ELISA) commercial CE-IVD kits based on availability from the reference laboratory. Cut-off values for low IgG avidity were <20% up to 2012, and <40% from 2013 onwards. An intermediate (“grey zone”) avidity index was defined as 20–80% until 2012 and 40–60% after 2012. Maternal viremia was detected by real-time PCR using in-house designed primers targeting the *UL55* gene and an in-house fluorescence resonance energy transfer (FRET)-based protocol with a cut-off value of 390 copies/mL [10]. The gestational age at diagnosis was considered the gestational week at which maternal primary CMV infection was diagnosed.

CMV HIG was delivered through the National Medicinal Agency procedure for the approval and administration of off-label drugs. The product used was Megalotect^®^ (Human cytomegalovirus immunoglobulin Biotest Pharma GmbH, Dreieich, Germany, 100 U/mL), given intravenously at a dose of 200 U/kg of actual body weight. The frequency of administration was initially every 4 weeks, while from January 2019 until July 2022, after evidence of favorable outcomes with biweekly administration emerged, the frequency was intensified to every 2 weeks.

CMV HIG was administered until amniocentesis was performed to assess vertical transmission with PCR testing of the amniotic fluid. Obstetric ultrasound and amniocentesis were offered to all women on the 21st gestational week. In cases of a negative PCR test after amniocentesis, HIG infusions were discontinued and the primary outcome was assessed with PCR testing of the newborns’ urine for CMV DNA within the first 10 days of life.

In cases of a positive PCR for CMV DNA in the amniotic fluid, termination of pregnancy was discussed when central nervous system (CNS) fetal abnormalities were detected by ultrasound. When a positive amniotic fluid for CMV DNA was accompanied by a normal ultrasound or extracranial findings, monthly HIG was offered until delivery as a treatment strategy [11], and follow-up included biweekly obstetric sonography and magnetic resonance imaging (MRI) of the fetus in gestational week 30.

If a woman declined amniocentesis, the continuation of monthly HIG until delivery was offered, following a strategy of “worst case scenario” (as if the amniotic fluid was positive). Every newborn was tested for CMV viruria. Newborns with positive PCR for CMV DNA were referred to a specialized pediatric infectious diseases team for evaluation, including a full blood count and biochemistry examination, quantitative PCR for CMV DNAemia, CNS imaging (ultrasound and/or brain MRI), abdominal ultrasonography, and a comprehensive eye and hearing examination (followed throughout the first 5 years of life, if initially normal).

We tried to address potential sources of bias with sensitivity analyses, examining the transmission outcome in the subgroups of women starting treatment during the first versus second trimester of pregnancy, and in those receiving biweekly vs. monthly infusions. Statistical analysis was performed with SPSS (IBM SPSS Statistics for Windows, v. 23.0. IBM Corp.: Armonk, NY, USA) and Microsoft Office Excel 2013. Dichotomous variables are presented as percentages, whereas normal continuous variables are presented as mean ± standard deviation, and those with nonparametric distribution as median (interquartile range). Dichotomous variables were compared with Chi square and continuous variables with Mann–Whitney or *t*-test. Missing values were not inferred. The threshold of statistical significance was set as *p*-value < 0.05 for all analyses. Univariable and multivariable logistic regression was performed to identify variables associated with vertical CMV transmission. Variables with *p* < 0.05 in univariable analysis were included in the multivariable model and those with *p* < 0.05 were retained until the final stage of the model. All statistical tests were two-tailed. 

## 3. Results

From an initial population of 533 pregnant women referred to our outpatient clinic for probable primary CMV infection based on positive CMV serology, 155 had a nonprimary infection (high IgG avidity testing ≤14th gestational week), and 95 additional women had IgG avidity with intermediate titers (“grey zone”) that were followed up as a separate cohort [12]. Among 279 women with maternal primary infection (MPI), 113 fulfilled the criteria for first trimester MPI and received HIG for congenital CMV prevention. In 112 women, a diagnosis was made based on positive IgG, IgM and a low IgG avidity. One woman was diagnosed by seroconversion. Six women suffered a miscarriage before amniocentesis was performed, leading to a population of 107 women evaluable for outcome. A detailed flowchart of the total cohort can be seen in Appendix A. 

### 3.1. Baseline Characteristics (Table 1)

The mean age of our cohort was 31.6 years, with 63.5% being nulliparous. Median gestational age at diagnosis was nine (8–11) and, at start of treatment, 13 (10–15) gestational weeks. Time from diagnosis to first HIG dose was at a median of three (2–4) weeks. Testing for CMV DNA with PCR in maternal blood upon presentation was conducted in 77 women, with a 35.1% positivity rate (27/77). From 21 with available quantitative measurement, the median viral load was 2320 (1000–7875) copies/mL. The initiation of HIG treatment more than two weeks after being diagnosed was recorded in 54.7% of our cohort, with 24.5% receiving the first HIG dose more than four weeks after their diagnosis. Thirty-six women (33.6%) diagnosed during the first trimester initiated HIG in the second trimester of their pregnancy (>14th gestational week). Median number of HIG doses was five (4–6). Seventy women received HIG every 4 weeks, while in 37 HIG was administered biweekly. During and immediately after HIG infusion, the women were monitored for allergic reactions and presence of headache. No infusion-related or obstetric adverse reactions were noted with the use of HIG in our cohort.

**Table 1 viruses-16-01241-t001:** Baseline patient characteristics, pregnancy outcomes and transmission rate (*n* = 107).

Baseline Characteristics	Values
Age, mean (SD)	31.6 (4.9)
Gestational week at diagnosis, median (interquartile range)	9 (8–11)
Gestational week at first HIG dose, median (interquartile range)	13 (10–15)
Time from diagnosis to first HIG dose (weeks), median (interquartile range)	3 (2–4)
Diagnosis to first HIG dose, *n* (%)	
≤2 weeks	48 (45.3)
>2 weeks	58 (54.7)
Diagnosis to first HIG dose, *n* (%)	
≤4 weeks	80 (75.5)
>4 weeks	26 (24.5)
Number of HIG doses, median (interquartile range)	5 (4–6)
Maternal viremia, *n/N* (%)	27/77 (35.1)
Amniocentesis, *n* (%)	89 (83.2)
Positive PCR amniotic fluid, *n/N* (%)	29/89 (32.6)
Termination of pregnancy with positive amniotic fluid, *n* (%)	10 (9.3)
Newborn urine PCR positivity, *n/N* (%)	22/96 (22.9)
Transmission rate, *n* (%)	32 (29.9)

HIG: hyperimmunoglobulin, PCR: polymerase chain reaction, SD: standard deviation.

### 3.2. Pregnancy Outcome and Transmission Rate

Amniocentesis was performed in 83.2% (89/107) of the total cohort (Table 1). Positive amniotic fluid PCR was found in 29 out of 89 (32.6%) pregnancies (4-weekly vs. biweekly HIG administration; 15/56, 26.8% vs. 14/33, 42.4%). One woman suffered a fetal loss after a negative amniocentesis and 10 pregnancies were terminated because of the detection of CMV DNA in amniotic fluid; of those, six had a normal obstetric ultrasound, three had CNS sonographic findings suggestive of cCMV embryopathy, while one woman was not evaluated. 

Nineteen women continued their pregnancy after the detection of CMV DNA in amniotic fluid. All of them had a normal obstetric ultrasound at the time of amniocentesis. Monthly HIG was continued at a dose of 200 IU/kg. Repeat monthly ultrasounds until delivery and fetal MRI at gestation week 30 did not reveal abnormal findings. CMV DNA in the newborns’ urine was positive in 18 out of 19. 

Eighteen women declined amniocentesis and 4-weekly 200 IU/kg HIG was offered until delivery. At gestation week 21, only one had an abnormal obstetric ultrasound (hyperechogenic bowel), with this finding remaining stable until delivery. Follow-up ultrasounds and fetal MRI for the remaining 17 women did not detect any anomalies. All newborns were assessed for CMV viruria. Four out of the 18 neonates were born with cCMV and all were asymptomatic at birth. The child with hyperechogenic bowel during gestation was negative for urine CMV DNA. 

From 96 children born, CMV DNA was detected in 22 of the newborns’ urine, for a total transmission rate of 29.9% (total cohort 32/107; 4-weekly vs. biweekly HIG administration; 18/70, 25.7% vs. 14/37, 37.8%). Only one neonate born with cCMV had symptomatic infection (4.5%). All congenitally infected children are under active follow-up from a specialized team of pediatric infectious diseases clinicians. The protocol of management and transmission outcomes can be seen in Figure 1. 

### 3.3. Maternal Viremia and Transmission Outcome

Among several variables tested, maternal viremia was the only factor which was significantly associated with fetal transmission in the total cohort (61.9% vs. 25%, *p* = 0.003) (Table 2). After a multivariate logistic regression analysis of factors associated with fetal transmission, maternal viremia remained positively associated with transmission rate after adjusting for age and receiving 4-weekly or biweekly HIG (OR = 4.62, 95% CI 1.55–13.74) (Table 3). 

### 3.4. Monthly vs. Biweekly HIG Administration 

Women receiving 4-weekly and biweekly HIG did not have significant differences in their baseline characteristics. Median gestational age at diagnosis for the 4-weekly group was 9 (8–11) and, for the biweekly group, 8 (7–11) gestational weeks. Both groups started HIG treatment at a median gestational age of 13 (10–15) weeks, and time from diagnosis to HIG treatment was at a median of three (2–5) weeks for both groups.

The transmission rate did not differ significantly between women who received 4-weekly compared to biweekly HIG, despite a trend for lower transmission rate in the 4-weekly-treated subgroup (25.7% vs. 37.8%, *p =* 0.193). Similarly, in the sub-analysis of women starting HIG treatment during the first trimester, a numerically lower transmission rate was noted in the 4-weekly-treated subgroup (20.5% vs. 40.7%, *p =* 0.065) (Table 4). 

### 3.5. Start of HIG Treatment during the First Trimester

In the subgroup of women starting HIG treatment during the first trimester, neither baseline characteristics nor maternal viremia rates were associated with fetal transmission (Appendix A). In a sub-analysis comparing women starting HIG treatment during the first or second trimester, the former were diagnosed (median gestational week, 8 [7–10] vs. 11 [9–13]) and started HIG treatment earlier (median gestational week, 11 [9–13] vs. 16 [15–17]), and received more HIG doses (median doses, 5 [4–7] vs. 3.5 [3–5]). The transmission rates between the two subgroups were not significantly different (first HIG dose in first vs. second trimester; 20/71, 28.2% vs. 12/36, 33.3%; *p =* 0.58). (Appendix A).

## 4. Discussion

Total transmission rate with use of CMV-specific HIG was 29.9% (32/107) in this multiyear retrospective analysis of first trimester primary CMV infections. The greatest benefit was observed in women with MPI starting treatment during the first trimester and receiving 4-weekly infusions, leading to a 44% reduction compared to the pooled transmission rates from a recent meta-analysis [6].

Pre-treatment maternal viremia was positively associated with fetal transmission after multivariate adjustment, in agreement with previous findings showing a 1.8-fold increase in fetal transmission (AOR, 3.0; *p =* 0.002) [13]. This emerging association suggests that CMV viremia could be included in the assessment of women with a primary infection as a high-risk marker of fetal transmission, and inform prevention strategies accordingly.

Our initial approach of monthly administration was based on early observational data showing a protective effect of 4-weekly CMV HIG [14]. Two subsequent randomized studies, however, using the same dose and frequency, failed to demonstrate this benefit [15,16]. Revello et al. found a lower transmission rate, although not significantly different (30% vs. 44%, *p =* 0.13), while Hughes et al. ceased their study due to futility, as an interim analysis showed no significant difference between treatment arms (relative risk, 1.17 [95% CI, 0.80–1.72]; *p =* 0.42). 

Compared to these randomized trials, we used a higher dose of 200 IU/kg, extrapolating from HIG treatment studies [11], as a contingency plan for the possibility of vertically transmitted CMV at presentation. Another important difference was gestational age at diagnosis, with randomized studies including women until late in the second trimester of pregnancy. In our cohort, the group of 4-weekly HIG administration had a transmission rate of 25.7%, a modest decrease from the meta-analysis reported rate of 36.8%. The potential significance of early gestational age at start of treatment could explain the even lower transmission rate of 20.5% in women that received the first HIG dose before gestation week 14.

In 2019, after initial evidence of a substantial decrease in transmission rates with biweekly HIG [17], we decided to change our practice and intensify the frequency of administration. Subsequent data by Kagan et al. showed a 6.5% transmission rate with biweekly 200 IU/kg HIG in recent first-trimester primary infections treated within one week after diagnosis [18]. In contrast, a prolonged delay of three weeks from diagnosis to treatment was observed in our study, and the transmission rate of the biweekly group was comparable to historical untreated cohorts. Another study, however, with intervals from diagnosis to treatment of less than a week, failed to detect any efficacy with biweekly administration, with transmission rates similar to natural infection and notably higher than monthly-treated women (42.9% vs. 26.7%; *p =* 0.448) [19]. An important consideration in the successful biweekly study was the use of seronegativity to anti-gB2-IgG to detect recent infections [18]. As anti-rec-gB2-IgG seropositivity indicates the emergence of CMV neutralizing antibodies and thus an earliest infection, its use as an exclusion criterion potentially selected women without cCMV at presentation. 

Compared to the 4-weekly-treated group, biweekly HIG in our cohort led to a higher, although not significant, transmission rate, despite similar baseline characteristics and time delay from diagnosis to treatment. As biweekly administration was adopted during the coronavirus disease 2019 pandemic, the implemented restrictions and public health measures could have been a confounding factor, leading to an unanticipated small sample size and limiting the power of our results [20].

Both CMV HIG and oral valacyclovir have shown potential benefit as in utero treatment strategies [11,21]. However, the use of historical cohorts as a comparator has been deemed to confer a high risk of selection bias and residual confounding [22]. As the evidence for the effective treatment of a confirmed CMV infection during pregnancy remained scarce, we decided to offer 4-weekly HIG administration until delivery to women with a positive amniocentesis and those that declined amniocentesis. An important prerequisite was the absence of CNS abnormalities on baseline obstetric ultrasound, in which case the termination of pregnancy was suggested. 

Women treated with CMV HIG for confirmed cCMV did not have any new cranial or extracranial anomalies on follow-up ultrasound and fetal MRI scans, and a low rate of symptomatic neonates was observed. Of note, these findings are in accordance with evidence from larger studies, suggesting that in cases of confirmed fetal infection normal serial prenatal imaging is associated with better postnatal outcomes [23]. 

No adverse obstetric reactions were recorded with HIG use; in contrast to previous HIG studies, where an increased number of complications, including preterm births, were noted [15].

The main strengths of this study are its well-defined intervention among women with documented first-trimester MPI. The paradigm shift to biweekly HIG treatment allowed us to compare transmission outcomes between the two dosing regimens. As a retrospective study, mirroring practice changes based on evolving literature, certain limitations need to be acknowledged. Treatment was offered to all women with evidence of infection, irrespective of time from diagnosis to consultation; thus, we could not exclude the possibility of HIG being offered after vertical transmission has already occurred. Furthermore, as maternal viremia was not a prerequisite for the diagnosis of a primary CMV infection, it was not consistently evaluated, leading to missing data. Lastly, the small sample size, especially of the biweekly group, limited the power of our study to identify significant differences in transmission rates.

## 5. Conclusions

In the past 20 years, observational and randomized studies examining the effect of HIG on cCMV prevention have reported discordant results with different dosage regimens, frequency of administration, and gestational age at enrollment. Recently published consensus recommendations favored oral valacyclovir for first-trimester primary infections and recommended against the use of monthly 100 IU/kg HIG, stating that biweekly 200 IU/kg HIG may be considered in cases of very recent infections [9]. 

Our retrospective data showed that, during a time with no proven options and high rates of pregnancy termination, CMV HIG was a safe, modestly effective strategy for cCMV prevention after primary infections. As HIG is a costly option with unclear benefits, and as the effectiveness of oral antiviral treatments emerge, its future role remains elusive. Evidence of maternal viremia’s association with vertical transmission could inform decisions in the initial assessment of pregnant women.

## Figures and Tables

**Figure 1 viruses-16-01241-f001:**
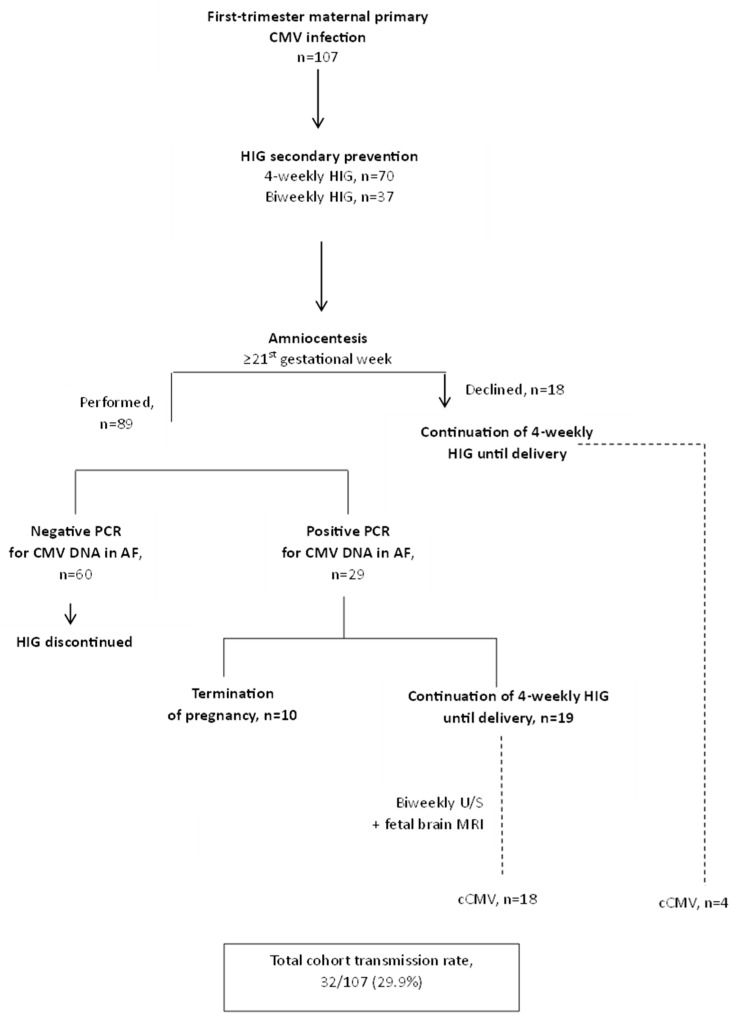
Protocol of management and transmission outcomes. AF: amniotic fluid, cCMV: congenital cytomegalovirus, CMV: cytomegalovirus, DNA: deoxyribonucleic acid, HIG: hyperimmune globulin, MRI: magnetic resonance imaging, *n*: number, PCR: polymerase chain reaction, U/S: ultrasound.

**Table 2 viruses-16-01241-t002:** Association of transmission outcome with baseline characteristics (*n* = 107).

	Transmission (−), *n* = 75	Transmission (+), *n* = 32	*p* Value
Age, mean (SD)	31.1 (4.9)	32.9 (4.9)	0.09
Gestational week at diagnosis, median (interquartile range)	9 (8–11)	9 (8–11)	0.76
Gestational week at first HIG dose, median (interquartile range)	13 (10–15)	12.5 (10–15)	0.97
Time from diagnosis to first HIG dose (weeks), median (interquartile range)	3 (2–4)	3 (2–5)	0.96
Time from diagnosis to first HIG dose, *n*/*N* (%)			0.68
≤2 weeks	33 (44)	15/31 (48.4)	
>2 weeks	42 (56)	16/31 (51.6)	
Time from diagnosis to first HIG dose, *n*/*N* (%)			0.76
≤4 weeks	46 (61.3)	20/31 (64.5)	
>4 weeks	29 (38.7)	11/31 (35.5)	
Number of HIG doses, median (interquartile range)	5 (4–6)	5 (3.25–7)	0.38
Maternal viremia, *n/N* (%)	14/56 (25)	13/21 (61.9)	0.003

Transmission (−): No vertical transmission. Transmission (+): Vertical transmission as defined in Section 2. HIG: hyperimmunoglobulin, *n*: number, SD: standard deviation.

**Table 3 viruses-16-01241-t003:** Multivariate logistic regression analysis of factors associated with fetal transmission.

Variable	OR (95% CI)	*p* Value
Age	1.08 (0.96–1.21)	0.23
Interval of HIG administration	1.35 (0.38–4.83)	0.65
Maternal viremia	4.62 (1.55–13.74)	0.006

CI: confidence interval, HIG: hyperimmunoglobulin, OR: odds ratio.

**Table 4 viruses-16-01241-t004:** Comparison of 4-weekly vs. biweekly HIG administration and transmission outcomes.

Total Cohort (*n* = 107)			*p* Value
	4-weekly HIG administration (*n* = 70)	Biweekly HIG administration (*n* = 37)	
Transmission rate, *n* (%)	18 (25.7)	14 (37.8)	0.193
**Start of HIG Treatment in the First Trimester Subgroup (*n* = 71)**			
	4-weekly HIG administration (*n* = 44)	Biweekly HIG administration (*n* = 27)	
Transmission rate, *n* (%)	9 (20.5)	11 (40.7)	0.065

HIG: hyperimmunoglobulin, *n*: number.

## Data Availability

The raw data supporting the conclusions of this article will be made available by the authors on request.

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
