# Peer review of "Cytomegalovirus-Specific Hyperimmune Immunoglobulin Administration for Secondary Prevention after First-Trimester Maternal Primary Infection: A 13-Year Single-Center Cohort Study"

_viruses, 2024, doi:10.3390/v16081241_

Round 1
Reviewer 1 Report
Comments and Suggestions for Authors
Authors performed a retrospective analysis of 200 IU/kg CMV-specific hyperimmune globulin use (HIG) in first-trimester maternal primary infections for congenital infection prevention as a 13-year cohort study. Importantly, they found maternal viremia was the only significant factor associated with fetal transmission. Transmission rate (29.9%) was not significantly different whether HIG was started during the first or second trimester, or between monthly and biweekly subgroups. Although this study did not confirm preventive efficacy of HIG against cCMV in women with maternal CMV primary infections, the study results include very important information for us where the efficacy and safety of VACV is still skeptical. Minor revision is needed.
Abstract
(32/107; 10 pregnancy terminations with positive amniocentesis, 18/19 completed pregnancies with positive amniocentesis and 4/18 declining amniocentesis)
need to be changed to
(32/107; 10 pregnancy terminations with positive amniocentesis, 18 completed pregnancies with positive amniocentesis and 4 declining amniocentesis)
Figure S1 and Line 130-134
How many women with seroconversion and positive IgM plus low avidity in n=279 and n=107?
Discussion
Because reduction was not proved,
CMV-specific HIG was associated with a modest reduction of fetal transmission in this multiyear retrospective analysis of first trimester primary CMV infections.
have to be corrected to
Total transmission rate with use of CMV-specific HIG was 29.9% (32/107) in this multiyear retrospective analysis of first trimester primary CMV infections.
Author Response
Comments 1: Authors performed a retrospective analysis of 200 IU/kg CMV-specific hyperimmune globulin use (HIG) in first-trimester maternal primary infections for congenital infection prevention as a 13-year cohort study. Importantly, they found maternal viremia was the only significant factor associated with fetal transmission. Transmission rate (29.9%) was not significantly different whether HIG was started during the first or second trimester, or between monthly and biweekly subgroups. Although this study did not confirm preventive efficacy of HIG against cCMV in women with maternal CMV primary infections, the study results include very important information for us where the efficacy and safety of VACV is still skeptical. Minor revision is needed.
Response 1: We thank the reviewer for the time taken to critically appraise and review our manuscript. We agree that our single-center multiyear analysis could not confirm the preventive efficacy of HIG, but hope it will add to the literature regarding the options for cCMV prevention after primary maternal CMV infection.
Comments 2: Abstract
(32/107; 10 pregnancy terminations with positive amniocentesis, 18/19 completed pregnancies with positive amniocentesis and 4/18 declining amniocentesis)
need to be changed to
(32/107; 10 pregnancy terminations with positive amniocentesis, 18 completed pregnancies with positive amniocentesis and 4 declining amniocentesis)
Response 2: We thank the reviewer for their insightful comment. We have changed this sentence in the Abstract (page 1, lines 24-25) and now reads: “(32/107; 10 pregnancy terminations with positive amniocentesis, 18 completed pregnancies with positive amniocentesis and 4 declining amniocentesis).”
Comments 3: Figure S1 and Line 130-134
How many women with seroconversion and positive IgM plus low avidity in n=279 and n=107?
Response 3: We thank the reviewer for their insightful comment. Unfortunately, from the 279 women diagnosed with primary CMV infection we have available data for 199. From these, 196 out of 199 women were diagnosed with a positive IgM plus a low IgG avidity and only 3 were diagnosed by seroconversion. In women with first-trimester primary CMV infection (n=113), 112 had a positive IgM plus low IgG avidity and only 1 woman was diagnosed through seroconversion. This information has been added in lines 135-136, which now read: “In 112 women, diagnosis was made based on positive IgG, IgM and a low IgG avidity. One woman was diagnosed by seroconversion.”
Comments 4: Discussion
Because reduction was not proved,
CMV-specific HIG was associated with a modest reduction of fetal transmission in this multiyear retrospective analysis of first trimester primary CMV infections.
have to be corrected to
Total transmission rate with use of CMV-specific HIG was 29.9% (32/107) in this multiyear retrospective analysis of first trimester primary CMV infections.
Response 4: We thank the reviewer for their valuable comment. We agree that this was not a controlled trial a correction should be made. The corrected statement (page 7, lines 221-222) now reads: “Total transmission rate with use of CMV-specific HIG was 29.9% (32/107) in this multiyear retrospective analysis of first trimester primary CMV infections.”
Reviewer 2 Report
Comments and Suggestions for Authors
The manuscript by Karofylakis et al., describes a retrospective, observational study focusing on the value of the application of cytomegalovirus hyperimmune globulin (HIG) to prevent congenital human cytomegalovirus infection (cCMV). Vertical transmission of human cytomegalovirus (HCMV) in the first trimester of pregnancy is associated with the risk of cCMV and with considerable disease burden in the offspring. The application of HIG has been proposed as a therapeutic means to prevent cCMV following proven primary HCMV infection of the mother. However, available studies provided controversial results about the value of this kind of intervention. This work reports a single-center experience addressing the correlation of the application of HIG in the first trimester of pregnancy with the postnatal detection of HCMV infection of the child as an endpoint. The study adds to our understanding of the role of HIG in the prevention of cCMV, though it can again not provide a conclusive answer about the impact of this intervention on the occurrence of cCMV and particularly on the outcome of cCMV with respect to clinical symptoms and sequelae.
1. A major issue about the use of HCMV-specific HIG is the quality of the product, as it is usually prepared as a pool of human sera. These may vary in the content of protective, antiviral antibodies. It would be helpful, if some information on the criteria for selection of the donors by the company could be provided.
2. It is stated in the Materials and Methods section, that the product MegalotectR was used. However, Biotest Pharma provides the product CytotectR. Is MegalotectR identical to CytotectR?
3. An Issue with the study is the variable time from diagnosis of primary HCMV infection in the mother to the initiation of HIG application. This is a limitation of the study, as rightfully discussed by the authors.
4. It appears confusing that there was a trend to a lower transmission rate in cases where HIG was applied on a monthly basis. The concurrent SARS-CoV-2 pandemic is acknowledged by the authors as resulting in a small sample size. Could it be that the restrictions implemented could be as such a confounding factor? E.g. were there documented COVID-19 cases in that group?
Author Response
Comments 1: The manuscript by Karofylakis et al., describes a retrospective, observational study focusing on the value of the application of cytomegalovirus hyperimmune globulin (HIG) to prevent congenital human cytomegalovirus infection (cCMV). Vertical transmission of human cytomegalovirus (HCMV) in the first trimester of pregnancy is associated with the risk of cCMV and with considerable disease burden in the offspring. The application of HIG has been proposed as a therapeutic means to prevent cCMV following proven primary HCMV infection of the mother. However, available studies provided controversial results about the value of this kind of intervention. This work reports a single-center experience addressing the correlation of the application of HIG in the first trimester of pregnancy with the postnatal detection of HCMV infection of the child as an endpoint. The study adds to our understanding of the role of HIG in the prevention of cCMV, though it can again not provide a conclusive answer about the impact of this intervention on the occurrence of cCMV and particularly on the outcome of cCMV with respect to clinical symptoms and sequelae.
Response 1: We thank the reviewer for the time taken to critically appraise and review our manuscript. We agree that our retrospective analysis could not provide a conclusive answer about the role of HIG in cCMV prevention, although we hope it will add to the literature regarding cCMV prevention options.
Comments 2: 1. A major issue about the use of HCMV-specific HIG is the quality of the product, as it is usually prepared as a pool of human sera. These may vary in the content of protective, antiviral antibodies. It would be helpful, if some information on the criteria for selection of the donors by the company could be provided.
Response 2: We thank the reviewer for their insightful comment. Unfortunately, we do not have any specific information about the criteria for selection of the donors by the company. According to the Summary of Product Characteristics, Megalotect® is prepared from plasma of donors who have high antibody titers against cytomegalovirus. Furthermore, the product’s SPC states that 1 ml solution contains 100 U antibodies against cytomegalovirus, where a unit is defined by the Paul Ehrlich Institute Standard (https://www.pei.de/DE/arzneimittel/antikoerper/immunoglobulinpraeparate/immunoglublinpraeparate-node.html;jsessionid=29DBBB9C2CE956CF6F9031BC2C19E0B3.intranet231).
Comments 3: 2. It is stated in the Materials and Methods section, that the product MegalotectR was used. However, Biotest Pharma provides the product CytotectR. Is MegalotectR identical to CytotectR?
Response 3: We thank the reviewer for their insightful comment. Yes, Cytotect® is identical to Megalotect®. Cytotect® is authorized in Greece under the name Megalotect®.
Comments 4: 3. An Issue with the study is the variable time from diagnosis of primary HCMV infection in the mother to the initiation of HIG application. This is a limitation of the study, as rightfully discussed by the authors.
Response 4: We thank the reviewer for their valuable comment. We agree that the variability in time from diagnosis of primary CMV infection to HIG initiation is a significant limitation and as such has been stated in the Discussion section.
Comments 5: 4. It appears confusing that there was a trend to a lower transmission rate in cases where HIG was applied on a monthly basis. The concurrent SARS-CoV-2 pandemic is acknowledged by the authors as resulting in a small sample size. Could it be that the restrictions implemented could be as such a confounding factor? E.g. were there documented COVID-19 cases in that group?
Response 5: We thank the reviewer for their insightful comment. The reviewer is correct to consider that the implemented restrictions could be a confounding factor to our study, and as such has been acknowledged in the limitations. We believe that the closure of nurseries and the widespread use of face masks led to fewer cases of primary cytomegalovirus infections in pregnant women. Regarding Covid-19 cases in our biweekly group, unfortunately this information was not systematically documented so we are unable to provide this information. However, according to our records, no infusion appointments were missed due to Covid-19 infection in women receiving biweekly HIG. We have amended lines 266-269, which now read: “As biweekly administration was adopted during the coronavirus disease 2019 pandemic, the implemented restrictions and public health measures could have been a confounding factor, leading to an unanticipated small sample size and limiting the power of our results”